# Glutaraldehyde Crosslinked High Content of Amylose/Polyvinyl Alcohol Blend Films with Potent Tensile Strength and Young’s Modulus

**DOI:** 10.3390/polym14245550

**Published:** 2022-12-19

**Authors:** Xinqing Wang, Zhenhua Huang, Zhaoyang Niu, Fangping Chen, Changsheng Liu

**Affiliations:** 1Key Laboratory for Ultrafine Materials of Ministry of Education, School of Materials Science and Engineering, East China University of Science and Technology, Shanghai 200237, China; 2The State Key Laboratory of Bioreactor Engineering, East China University of Science and Technology, Shanghai 200237, China

**Keywords:** high content of amylose, amylose/PVA blend film, crosslinking, mechanical property, starch

## Abstract

In recent years, with the development of green environmental protection, starch film has become of interest due to the wide availability of sources, low price, and biodegradability. Amylose/polyvinyl alcohol (PVA) blend films crosslinked with different amounts of glutaraldehyde (GLU) were prepared by a solution casting method. The cross-linking degree, water sorption, tensile property, crystallization and section morphology of the films were examined. With the increase in glutaraldehyde concentration, the cross-linking degree of the blend film was improved. The wide-angle X-ray scattering (WAXS) result indicated that cross-linking hindered the crystallization of film. The section morphology of films was examined by scanning electron microscope (SEM). The results showed that the cross-linking degree of amylose film improved while the crystallinity decreased with the increase in glutaraldehyde content. Cross-linking had no obvious effect on the water sorption property of the blend films. The cross-linking modification significantly enhanced the tensile strength and Young’s modulus, while it reduced the elongation at break of the blend films. It was found that the film with 0.5 wt % glutaraldehyde possessed the best performance: the tensile strength increased by 115%, while the elongation at break decreased by 18% even at high relative humidity (RH) of 90% compared to non-crosslinked films. The developed amylose/PVA blend films have promising application prospects as agricultural mulch films and packaging materials.

## 1. Introduction

An increasing demand of plastic production and damage to the environment of petroleum-based plastic wastes are urging people to develop environment-friendly substitutes, which should be biodegradable, non-toxic and abundant. Various biopolymers, such as starch, cellulose, protein and chitosan, have been developed to replace synthetic polymers. As one of the most promising biopolymers, starch combines biodegradability, abundance, low cost, renewability and easy processing. Despite many advantages of starch, it is highly sensitive to water and has a low mechanical strength. It was reported that the mechanical strength of cassava starch films with 20% glycerol decreased from 21 to 2 MPa when the relative humidity (RH) increased from 32% to 90%. Meanwhile, the plasticizer enhanced the water sensitivity but reduced the mechanical properties [1]. In addition, mechanical properties often deteriorate when starch films are exposed to air owing to its water sorption property. All these deficiencies greatly limit the wide application of starch-based materials. Therefore, it is urgent to improve the water resistance and mechanical properties of starch-based materials.

Increasing the amylose content in starch is one of the effective methods to improve the mechanical properties and stability of starch films in a high RH environment [2,3,4]. Starch film with high amylose content helps inhibit the recrystallization and aging of materials. Meanwhile, the formation ability and its stability of film were enhanced [3,5]. When the amylose content increased from 0 to 100%, a continuous increase was observed in tensile strength (from 40 to 70 MPa) and elongations (from 4% to 6%) of starch films [6,7]. The amylose films with 20% glycerol showed a much higher elongation than those with low glycerol contents, while the amylopectin produces a very weak and nonflexible film [8]. Muscat et al. reported that the films with high amylose content exhibited higher tensile strengths, improved elasticity modulus and lower elongations at break than those films with low amylose content [9]. Compared with the amylopectin films, an amylose film is more stable in water. When immersing the amylose and amylopectin films in water, less than 3 wt % of amylose film was dissolved, while about 60 wt % amylopectin was dissolved. The main reason was that the linear chains were much closer to each other than amylopectin and possessed a higher extent of hydrogen bonds, which was in favor of the mechanical properties and water stability of starch films [10,11].

Chemical modification is another strategy to improve the mechanical properties and water resistance of starch. To change the intrinsic properties of native starch, such as weak water resistance, swelling, low tensile strength and toughness, the polysaccharides were modified via reactions with hydroxyl groups. Multifunctional molecules tend to be selected to react with starch to form intra- and inter-molecular bonds, so that the starch molecular chains can combine with each other and form the cross-linked structure. Native starch cross-linked by citric acid, epichlorohydrin, glutaraldehyde and malonic acid were studied [12,13,14,15,16]. For example, Liu et al. explored the effect of the cross-link agent glutaraldehyde on the properties of thermoplastic starch/PVA blend films [10]. The starch films cross-linked by glutaraldehyde were used as edible films with a rapid biodegradation [11]. Cross-link agents such as sodium trimetaphosphate and epichlorohydrin were successfully cross-linked with high amylose content (70%) [17,18,19]. The mechanical properties and water stability of starch/PVA blends were increased when starch molecules and PVA molecules were interconnected by covalent bonding after cross-linking modification [20]. Liu et al. revealed that cross-linking modification with sodium hexametaphosphate significantly reduced the moisture sensitivity and increased the tensile strength and Young’s modulus of the thermoplastic starch (TPS)/PVA blend films [21]. The mechanical properties of the starch-based film were improved with glycerol as a plasticizer and polyvinyl alcohol (PVA) as a second component [22,23,24]. Early in the 1940s, the preparation and characteristics of amylose films were studied in detail by Ivan A. Wolff [25]. However, up to now, the cross-linking modification of starch film high amylose content (>90 wt %) has rarely been investigated.

To solve the problems of high humidity sensitivity and low mechanical properties of starch films, the amylose was firstly prepared by the improved butanol precipitation method, and the amylose/polyvinyl alcohol blend films (APF) with high tensile strength and Young’s modulus were prepared by glutaraldehyde (GLU) cross-linking and solution-casting method. The introduction of a cross-linking agent forms a chemical bond between the starch and the hydroxyl group on the PVA and creates a three-dimensional network structure. The strength of the starch/PVA cross-linking network is related to the cross-linking density and the degree of crystallization. The increase in cross-linking density will improve the tensile strength. In addition, for gelatinized starch, the residual crystal structure acts as the reinforcement in the amorphous starch molecular chains, and the decrease in crystallinity may lead to the decrease in mechanical strength. Furthermore, the cross-linking degree, moisture absorption, RH sensitivity and mechanical properties of APF blend films with different concentrations of crosslink agent and under different relative humidities were investigated.

## 2. Materials and Methods

### 2.1. Materials

Amylose (Amylose > 95 wt %, Mw = 2 × 10^5^~5 × 10^5^) was obtained from the State Key Laboratory of Bioreactor Engineering (Shanghai, China). PVA (Mw = 4.4 × 10^5^), glycerol, alcohol, sodium chloride and 25 wt % glutaraldehyde solution were purchased from Shanghai Lingfeng chemical reagent co. LTD (Shanghai, China). Dimethyl sulfoxide (DMSO) was purchased from Aladdin industrial corporation (Shanghai, China).

### 2.2. Preparation of Amylose/PVA Blend Films

The preparation of crosslinked amylose/PVA blending film was described as follows. At first, amylose was prepared by an improved butanol-precipitation procedure [26]. In detail, amylose powder (3 g) was added to 10 mL alcohol and mixed for 10 min to improve the water solubility of powder. Alcohol here was used as a complexing agent to promote the solubility of amylose. Then, the aqueous dispersion of amylose was prepared by adding 90 mL of deionized water. Afterwards, the suspension was heated to boiling with continuous stirring and kept for 15 min to dissolve the amylose completely. Then, glycerol (0.6 g) and PVA (0.6 g) were added to the above solution, respectively. The solution was adjusted to pH = 5 with acetic acid. In order to obtain different cross-linking degrees, glutaraldehyde at the concentration of 0%, 0.25%, 0.5%, 1%, 1.5%, 2% and 5% (*w*/*w*) of the amylose were added in different reaction bathes, respectively. After being stirred for another 15 min, the amylose solution was filtered through cotton gauze and cast onto a polymethylmethacrylate (PMMA) sheet (15 × 20 cm^2^). After cooling at 25 °C for 40 min, the solution became milk white and gelatinous, and we kept them in an oven (DHG-9037A, Shanghai Huilu Scientific Instrument Co., Ltd., Shanghai, China) at 50 °C for three hours. When the solution was evaporated completely, the films with 40~50 μm in thickness were removed from PMMA sheets and stored in a humidity chamber (LHS-100CH, Shanghai Yiheng Scientific Instruments LTD, Shanghai, China) with RH = 50% which was obtained used sodium chloride saturated solution [2,3]. The amylose film without PVA was chosen as control and prepared by the same method. The compositions of APF were listed in Table 1.

### 2.3. Cross-Linking Degree

The cross-linking degree was indirectly determined by the weight loss of films before and after soaking in DMSO. First, all films (40 × 40 mm^2^) were dried to constant mass and weighed as m_0_. After immersion in 20 mL DMSO for 1 h, photographs were taken by a digital camera (Nikon D5000, Tokyo, Japan) to observe their gel fraction. After immersion for 24 h, the film was considered to reach solubility equilibrium. The insoluble part of films was filtered out by cotton gauze and weighed as m_s_, and the swollen part was dried to a constant mass and weighed as m_d_.

The swollen degree (SD) and solubility of the film in DMSO were used to characterize the cross-linking degree. The swollen degree and solubility were calculated according to Equations (1) and (2) [27].
SD = (m_s_ − m_d_)/m_d_(1)
Solubility (%) = (m_d_/m_0_) × 100(2)

### 2.4. Fourier Transform Infrared Spectroscopy (FTIR)

FTIR spectra of films were recorded on a Nicolet 5700 spectrometer (Thermo Fisher Scientific, Waltham, MA, USA). The wavelength range was 4000–1300 cm^−1^ and the scanning resolution was 2 cm^−1^. All the films were stored in desiccators at RH = 50% for 3 days before test. All the films were placed on a Miracle ATR accessory (Thermo Fisher Scientific, Waltham, MA, USA), and the spectra were recorded in the reflection mode.

### 2.5. Crystallinity

The crystallinity of the film dried in vacuum was analyzed by an X-ray polycrystalline diffract meter (XRD, D/max-2550 VB/PC, Hitachi LTD, Tokyo, Japan) with Cu-Kα1 radiation source (tube operating at 40 kV, 100 mA) in a continuous scan mode. The diffraction data were collected from 10° to 80°, with a step size of 0.02° at a scan speed of 3°/min. The relative crystallinity of film was determined by the ratio of the crystalline peak diffraction area to the total diffractogram area. XRD-peak-diffraction analysis was used to calculate the area of crystalline peaks and amorphous peaks by MDI jade5.0 according to Equation (3) [28,29].
Crystallinity (%) = S_(crystalline peaks)_/(S_(crystalline peaks)_ + S_(amorphous peaks)_) × 100 (3)

### 2.6. Water Vapor Sorption Isotherm

The water vapor sorption properties of film were measured by storing the samples at 25 °C and humidity chamber with RH = 25, 45, 54, 67, 75, 80, 90%. All samples were dried to a constant mass (w_1_) before being stored in the chamber. After equilibrium for 2 days, the final weighed mass of the film after equilibrium was recorded as w_2_. The water content of film (MC) was calculated by Equation (4).
MC (%) = (w_2_ − w_1_)/w_1_ × 100(4)

### 2.7. Mechanical Properties

The mechanical properties including tensile strength, Young’s modulus and elongation at break were measured by a universal testing machine (CMT2503, MTS System Corporation, Shanghai, China) according to the GB/T 1040.3-2006 test standard. The film is cut into dumbbell-shaped splines with a narrow neck width of 4 mm and thickness of 40~50 μm measured by a thickness gauge (CH-10-A, Shanghai Liu ling Instrument Factory, Shanghai, China) with a precision of 0.01 mm. All films were conditioned at 25 °C with RH of 25%, 50%, and 90% for two days before test. The mechanical properties were measured at a loading rate of 10 mm/min, and the original gauge length was 20 mm with a 5000 N load cell. For each film, at least five sets of samples are tested.

### 2.8. Morphology Observation

To evaluate the effect of crosslink agent, the test specimens were sputter-coated with gold for 45 s. The morphology of the films’ tensile failure section after mechanical test was observed by scanning electronic microscopy (SEM, Hitachi S-3400N, Hitachi, Japan) under 15 KV acceleration voltage. The micrographs of the samples were taken at 1000 × magnification.

### 2.9. Statistical Analysis

All the data were expressed as mean ± standard deviation. One-way ANOVA and test were used for statistical analysis using SPSS19.0 software (IBM SPSS Statistics 19.0, International Business Machines Corporation, New York, NY, USA), and * *p <* 0.05, ** *p <* 0.01, *** *p <* 0.001.

## 3. Results and Discussion

### 3.1. Cross-Linking Degree

The natural starch, retrograded starch, and other starch without cross-linking can be dissolved by DMSO, but starch is hard to dissolve after cross-linking [24,30,31]. In this manuscript, the solubility in DMSO and SD of cross-linking films was selected to represent the cross-linking degree.

Figure 1 shows the solubility of different blend films after immersion in DMSO for 1 h. APF/G0 was almost dissolved completely (Figure 1a), indicating that DMSO has good solubility for amylose with a linear chain and PVA. With the addition of glutaraldehyde, the profile and morphology of APF/G0.25, APF/G0.5 and APF/G1 were well maintained. The strip-like contour of films became more and more clear with the increase in glutaraldehyde content (Figure 1b–d). The results showed that amylose and the PVA molecule were cross-linked in the presence of glutaraldehyde. It was reported that under acidic conditions, two aldehyde groups on glutaraldehyde could be reacted with hydroxyl groups of amyloses and PVA, and the newly formed molecule was too large to be swelled and dissolved [32]. These chemical bonds formed a stable three-dimensional network structure, much stronger than hydrogen bonds, and enhanced the water resistance of amylose/PVA blend films.

Figure 2 shows that the SD increased while the solubility of film decreased quickly with the increase in the crosslink agent (glutaraldehyde) concentration. The result indicated that the cross-linking degree of the film was improved. When glutaraldehyde reached up to 2 wt %, the cross-linking degree was not obviously improved, which illustrated that the concentration of crosslink agent had reached the maximum. The value of SD and solubility of the films with different concentrations of glutaraldehyde are listed in Table 2. Although Figure 2 and Table 2 showed the effect of different concentrations of glutaraldehyde on the solubility and swelling in DMSO of the blend films, the maximum solubility at 0% glutaraldehyde was about 83.71%. That is to say, the remaining 16.31% insolubility may originate from the cross-linking role of glycerol.

### 3.2. FTIR Analysis

The FTIR spectra of the amylose/PVA blend films crosslinked with different glutaraldehyde concentrations are displayed in Figure 3. The strong and wide absorption band at about 3277 cm^−1^ was ascribed to the –OH stretching vibration of hydrogen bonds in the blend films, which was mainly corresponding to hydroxyls in PVA, glycerol, and amylose [33,34,35,36]. The characteristic absorptions at 2923 cm^−1^ were the asymmetric stretching vibrations of –CH_2_ groups in PVA, glycerol, and amylose [33,34,35,36].

A small peak at 1740 cm^−1^ was corresponding to the acetyl group in PVA due to its incomplete saponification [33]. The peaks at 1652 cm^−1^ and 1455 cm^−1^ are due to the hydroxyl group absorption peaks and symmetric bending of –CH_2_ in –CH_2_OH, respectively [33,34]. The peak near 1150 cm^−1^ is attributed to the stretching vibration of the C–OH bond in PVA, glycerol and amylose [33,34,35,36]. The strong and sharp peak at 996 cm^−1^ is attributed to C–O–C of α-1,4-glycosidic bonds in amylose [34]. The dominant difference in blend films was the intensity of the –OH peak. The intensity of –OH stretching vibration absorption had a decline as the cross-linking degree increased, which indicated the reduction in –OH in the cross-linking films.

### 3.3. Wide-Angle X-ray Scattering (WAXS) and the Film’s Crystallinity

Amylose film had a B-type crystallinity, whereas the amylopectin film was amorphous [37]. The XRD patterns of all films are shown in Figure 4, and the crystallinity degree of the films is listed in Table 3. The characteristic peak of PVA diffraction was at 2θ = 19.6° and the crystallinity of pure PVA was 41.5%, the AF without PVA and crosslink agent had the main X-ray diffraction at 2θ = 16.94° and 22.24°, which indicated the presence of B-type crystallinity [31,37]. There was a sharp peak at 19.6° in APF/G0.25, which was attributed to the ordered arrangement of PVA molecules in the blend film, and the characteristic peaks at 17.04° and 22.52° were corresponding to the amylose molecules. With the increase in crosslink agent concentration, the peaks at 2θ = 16.96°, 19.74° and 22.5° became broader and weaker, which indicated the crystallinity reduction in film, therefore a continuous transformation from the hypocrystalline to amorphous state. The result coincided with the literature by Rioux [30]. The covalent cross-linking of film hindered the free movement of molecule chains and crystal formation of film. As a result, the crystallinity of blending film decreased from 31.2% to 9.74% when the GLU was from 0 wt % to 5 wt %.

The peaks at 16.9° which were corresponding to the crystallinity of amylose molecules reduced as compared to the peaks at 19.7° and 22.4°. This result indicated that the mobility of some amylose chains was not affected by crosslink agents, and their free movement formed an ordered structure in the film. The other amylose chains and PVA molecules were much easier to react with GLU, which limited their mobility and hindered the crystallization.

### 3.4. Moisture Sorption Property

Figure 5 shows the water vapor sorption ability of all samples at different surrounding humidity. The water content of film increased with the increase in RH considerably, especially at RH > 75%. The result showed it was apt to absorb water at a high surrounding humidity for blend film. In addition, the increment of water vapor sorption was mainly due to the phase separation between glycerol and the amylose/PVA matrix when RH exceeded 75%. Before phase separation, the amylose/PVA matrix and glycerol were combined by a hydrogen bond, and the matrix was the main component to absorb water vapor, and the glycerol combined with water too after phase separation [38,39].

Figure 5 shows the sorption curves of different cross-linking degree films. The similar water sorption behavior is due to the following two factors: On one hand, the decrease in hydroxyl group content decreased the binding capacity of water with film. On the other hand, cross-linking made the film structure amorphous and loose, which was in favor of water sorption. When the concentration of the crosslink agent reached 5 wt %, the blend film possessed the highest water content, especially at the relative humidity of 90%. The result indicated that the structure of the film affected the water sorption of films.

### 3.5. Mechanical Properties

Table 4 shows the mechanical properties including tensile strength at maximum load, elongation at break and Young’s modulus of the amylose/PVA blend films with different cross-linking degree and relative humidity. As was reported in many studies in the literature, tensile properties were controlled by the relative humidity and the structure of starch-based film [15,30,40,41]. In this paper, the effect of relative humidity and cross-linking degree of the blend film on the mechanical properties was explored.

With the increase in crosslink agent content, tensile strength and Young’s modulus had a similar trend. At low contents of crosslink agent (0.25 and 0.5 wt %), the tensile properties of amylose/PVA blend films improved significantly, especially at the GLU content of 0.5 wt %. When the crosslink agent reached 0.5 wt %, the tensile strengths were 67.03 ± 8.20 MPa, 33.57 ± 4.34 MPa and 8.88 ± 2.72 MPa at RH = 25%, 50% and 90%, which increased by 48%, 42% and 115% compared to AF/G0, respectively. The result revealed that cross-linking modification enhanced the mechanical property of the blend films especially at a high relative humidity. A rigid network structure, formed by the cross-linking modification, improved the Young’s modulus and tensile strength while it decreased the toughness of the blend film. The further increment of cross-linking agent content in films showed a little reduction in tensile strengths and Young’s modulus. According to the result of X-ray diffraction, the high content of crosslink agent led to a lower crystallinity degree and a looser structure of film, which weakened the mechanical performance, especially the toughness.

Table 4 shows that the surrounding humidity had a great influence on the toughness of film. At RH = 25%, the elongations at break of all films were much lower than the groups at RH = 50% and 90%. As an effective plasticizer for amylose/PVA films, water promoted the mobility. When the water content reached a certain value, phase separation of the film occurred. In addition to the surrounding humidity, Table 4 also shows that the cross-linking degree affected the toughness of films. With the addition of GLU, the elongation of amylose/PVA blend films declined sharply. Therefore, cross-linking limited the motivation of molecule chains and made the films fragile. Generally speaking, the mechanical properties of amylose/PVA blend films have been greatly improved by cross-linking. Compared with commercial packaging bags and agricultural mulch films, glutaraldehyde crosslinked starch films possessed high tensile strength, Young’s modulus and excellent biodegradability [32,42].

### 3.6. Section Microstructure

Figure 6 shows the tensile failure section microstructure of the amylose film. The fracture surface on APF/G0 film without glutaraldehyde addition (Figure 6a) was smoother and denser than that on cross-linking films, and it even showed a brittle fracture. All blend films after cross-linking (Figure 6b–f) had a rough surface and scaly-like texture. When the glutaraldehyde content reached 5 wt %, the fracture surfaces became loose and porous (Figure 6f), which indicated cross-linking changed the film structure. The SEM observation was consistent with the result of the WAXS spectrum. With the increased content of glutaraldehyde in film, the molecule chains were crosslinked with each other and became amorphous. This decreased the crystallinity and made the fracture surfaces rough and uneven. The film structure also changed the mechanical property. Because the fracture surfaces became rough and scaly-like, the tensile strength of the film declined slightly. In addition, the film did not present brittle fracture any longer after the cross-linking process.

## 4. Conclusions

The amylose/PVA blend films crosslinked with different amounts of glutaraldehyde were prepared successfully by a solution cast method. The results showed that the modification of cross-linking for the amylose/PVA blend film improved the tensile strength, Young’s modulus and water resistance even at a high surrounding humidity significantly. The crystalline of the amylose/PVA blend films decreased to 9.7% and the stability improved with the increased addition of glutaraldehyde at high RH. In addition, compared to the films APF/G0 before modification, the tensile strength of APF/G 0.5 increased by 48%, 29% and 115% at RH = 25%, 50% and 90% humidity, respectively. The increase in strength of APF/G 0.5 was more pronounced at high humidity. The APF/G 0.5 blend films with 0.5 wt % GLU had the best mechanical performance with the tensile strength of 67.03 ± 8.20, Young’s modulus (MPa) of 16.66 ± 1.34 MPa and elongation at break of 77.45 ± 5.77%. The amylose/PVA blend films with high mechanical properties and potent water resistance is expected to be used as packaging materials and agricultural mulch films.

## Figures and Tables

**Figure 1 polymers-14-05550-f001:**
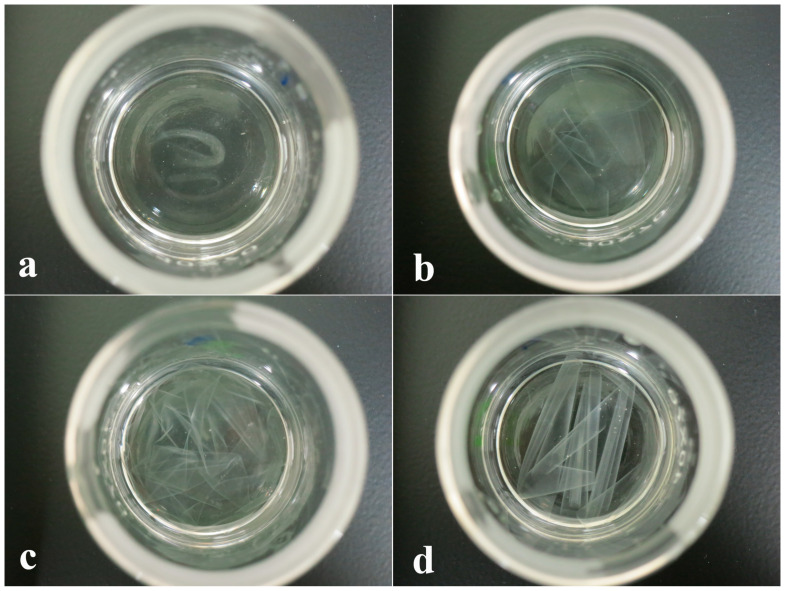
Solubility of films after immersion in DMSO for 1 h. (**a**) APF/G0, (**b**) APF/G0.25, (**c**) APF/G0.5, (**d**) APF/G1.

**Figure 2 polymers-14-05550-f002:**
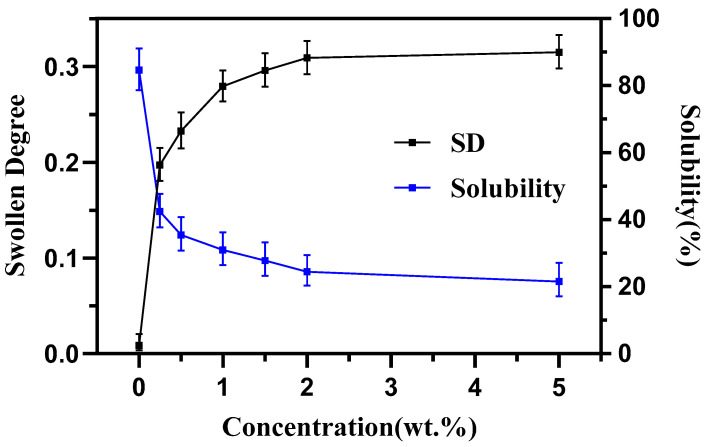
Swollen degree (SD) and solubility of cross-linking amylose/PVA blend films with different glutaraldehyde concentrations.

**Figure 3 polymers-14-05550-f003:**
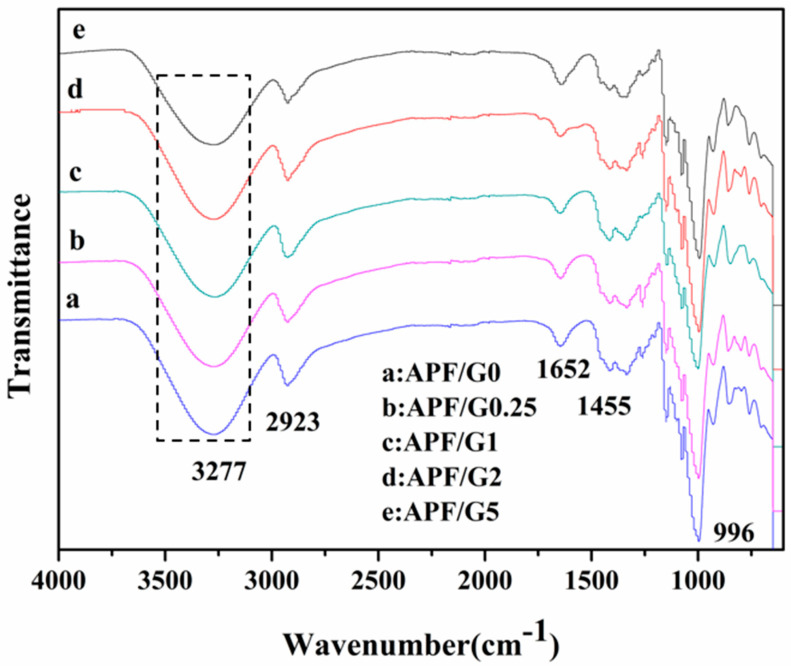
FTIR spectra of (a) APF/G0, (b) APF/G0.25, (c) APF/G1, (d) APF/G2 and (e) APF/G5.

**Figure 4 polymers-14-05550-f004:**
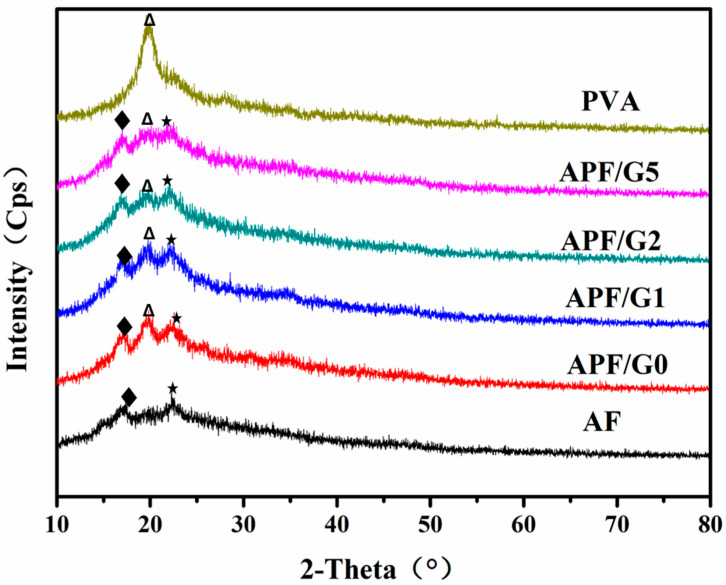
WAXS of APF/G (0, 1, 2 and 5), AF and pure PVA film.

**Figure 5 polymers-14-05550-f005:**
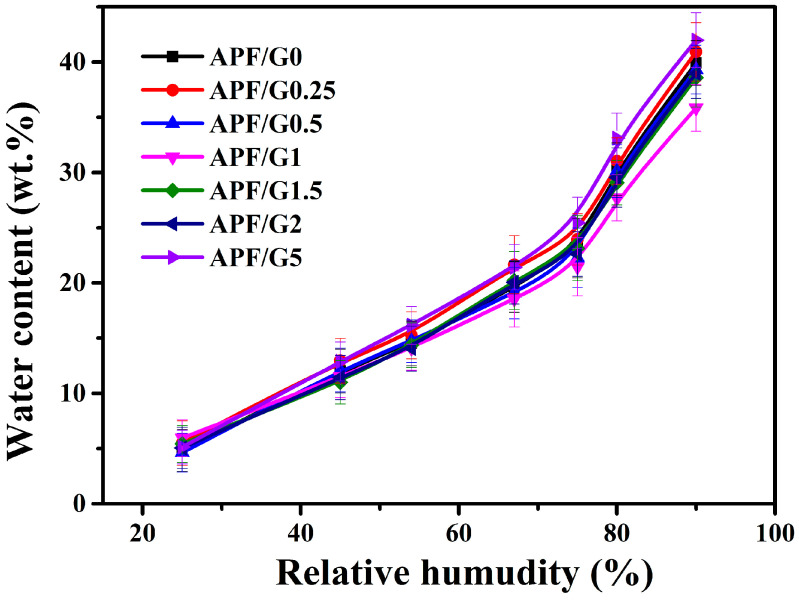
Water vapor sorption isotherms at 25 °C for different blending films.

**Figure 6 polymers-14-05550-f006:**
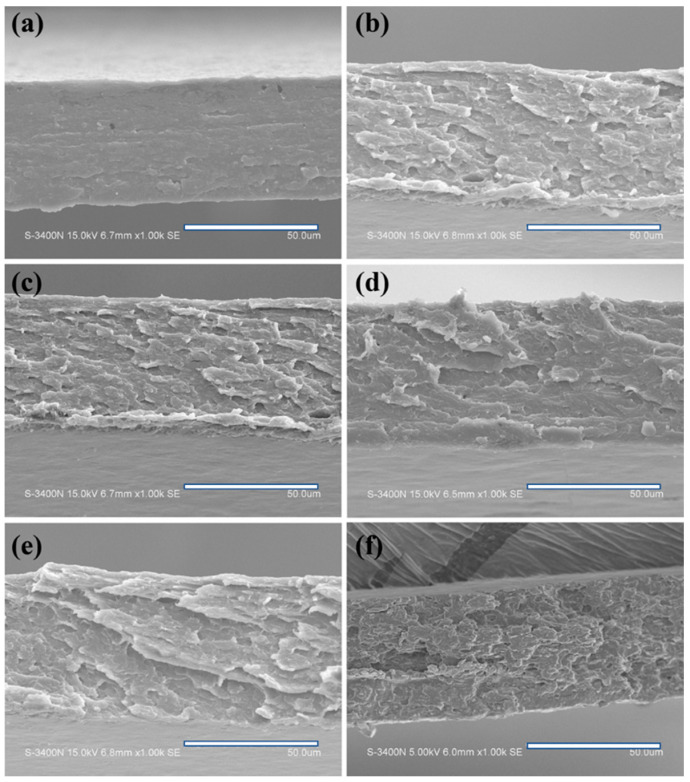
SEM images of stretching section of amylose/PVA blend films ((**a**) APF/G0, (**b**) APF/G0.5, (**c**) APF/G1, (**d**) APF/G1.5, (**e**) APF/G2 and (**f**) APF/G5), at 1000 × magnification.

**Table 1 polymers-14-05550-t001:** Compositions of amylose/PVA blend films.

Samples	Amylose (wt %)	PVA (wt %)	Glycerol (wt %)	GLU (wt %)
APF/G0	71.40	14.30	14.30	0
APF/G0.25	71.40	14.30	14.30	0.25
APF/G0.5	71.40	14.30	14.30	0.50
APF/G1	71.40	14.30	14.30	1.00
APF/G1.5	71.40	14.30	14.30	1.50
APF/G2	71.40	14.30	14.30	2.00
APF/G5	71.40	14.30	14.30	5.00
AF	83.31	0	16.70	0

**Table 2 polymers-14-05550-t002:** Swelling degree and solubility of amylose/PVA blend films in DMSO.

Samples	SD	Solubility (%)	Immersion for 1 h
APF/G0	0.02	83.71	Almost completely dissolved
APF/G0.25	0.20	42.19	Partly dissolved
APF/G0.5	0.24	33.69	Partly dissolved
APF/G1	0.28	29.33	Minimally dissolved
APF/G1.5	0.29	25.81	Minimally dissolved
APF/G2	0.30	23.73	Minimally dissolved
APF/G5	0.31	21.30	Minimally dissolved

**Table 3 polymers-14-05550-t003:** Relative crystallinity degree (%) of samples with different concentrations of cross-linking agent.

Samples	Relative Crystallinity Degree (%)
AF	14.1
APF/G0	31.0
APF/G0.25	29.7
APF/G0.5	28.6
APF/G1	27.8
APF/G1.5	23.1
APF/G2	15.6
APF/G5	9.74
PVA	41.5

**Table 4 polymers-14-05550-t004:** Mechanical properties of amylose/PVA blend films under different RH.

Mechanical Properties	RH (%)	Samples						
		APF/G0	APF/G 0.25	APF/G 0.5	APF/G 1	APF/G 1.5	APF/G 2	APF/G 5
Tensile strength (MPa)	25	45.28 ± 2.86	64.40 ± 5.37	67.03 ± 8.20	63.29 ± 4.32	61.15 ± 4.37	59.28 ± 3.97	57.20 ±3.91
	50	23.70 ± 1.54	31.28 ± 2.54	33.57 ± 4.34	30.58 ± 2.49	28.02 ± 2.42	27.83 ±1.98	27.25 ±1.40
	90	4.13 ± 2.86	7.64 ± 4.87	8.88 ± 2.72	6.69 ± 3.77	5.41 ± 5.10	4.93 ± 1.09	4.53 ± 1.19
Young’s modulus (MPa)	25	10.90 ± 1.14	17.39 ± 1.24	16.66 ± 1.34	16.05 ± 1.29	15.76 ± 1.13	15.85 ± 0.79	14.73 ± 1.41
	50	4.09 ± 0.44	5.08 ± 0.84	4.82 ± 1.24	4.37 ± 0.75	4.49 ± 0.82	4.49 ± 0.59	4.33 ± 0.68
	90	0.29 ± 0.04	0.96 ± 0.04	0.94 ± 0.05	0.75 ± 0.03	0.71 ± 0.03	0.61 ± 0.04	0.52 ± 0.04
Elongation at break (%)	25	31.84 ± 4.21	5.77 ± 1.8	11.24 ± 2.87	13.43 ±3.28	18.81 ± 4.87	14.77 ± 3.03	7.11 ± 2.34
	50	98.34 ± 8.23	74.52 ± 3.87	77.45 ± 5.77	68.52 ± 3.68	67.17 ± 3.27	65.80 ± 4.51	58.69 ± 3.79
	90	76.44 ± 6.37	55.45 ± 6.36	62.95 ± 8.36	53.25 ± 3.07	49.59 ± 2.68	41.47 ± 2.55	25.92 ± 2.17

## Data Availability

The author declares that all the data in the article are true and valid. If you need to quote, please indicate the source.

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
