# Peer review of "Glutaraldehyde Crosslinked High Content of Amylose/Polyvinyl Alcohol Blend Films with Potent Tensile Strength and Young’s Modulus"

_polymers, 2022, doi:10.3390/polym14245550_

Round 1

Reviewer 1 Report

The authors describe the formulation, characterization, and analysis of amylose-PVA-glycerol polymers containing increasing levels glutaraldehyde cross-linker.  There are many previous studies examining the effects of glutaraldehyde crosslinking on starch, PVA, and other polymers, however it appears that the unique aspect of this work is that the authors focus on a high amylose content.  Potential applications of food packaging and agricultural films are mentioned in the abstract and conclusion, however these are not discussed elsewhere nor are the material’s properties placed in context relative to the field.  Further, several of the results and discussion require expanded discussion, reconsideration of conclusions, and/or comparative analysis with other results to extract clearer and more impactful conclusions.  While interesting, this work is fairly incremental relative to recent literature and without major revision, does not represent a significant finding.  Therefore, major revisions are recommended prior to reconsideration for publication.  Please see specific comments below.

·         Proofread for minor corrections with English

·         abstract and conclusion are only mention of potential applications. need to compare the APF-Gx film properties to current commercial packaging materials and/or agriculture mulch films.  how do the mechanical properties compare? water uptake?

·         Page 2, lines 48-50: confusing sentences – rework

·         Page 2, lines 85-88: run-on sentence – correct

·         2.1 materials: add units to Mw of amylose

·         2.1 materials: use convention units for mw of PVA

·         2.3: recommend changing "solubility" to "Gel fraction", which is conventionally accepted term for the fraction remaining after extraction with appropriate solvent.

·         2.3: Describe how were samples dried. in vacuo? DMSO is difficult to completely remove

·         2.7: a 5000N load cell is a very high capacity load cell - what is the sensitivity of this? detection limit? typically resolution of load cells are on the order of 1/500 of load cell capacity. which in this case would be 10 N, a fairly high force considering the properties of the samples.

·         3.1, fig. 1: discussion of figure 1 is entirely subjective.  While pictures do clearly show dissolution at 0% cross-linking, too much is extrapolated from the others.

·         Page 5, lines 180-182: very confusing.  Is this sentence in reference to literature, or this work? add citation if to literature, or clarify where acid is introduced if not

·         Fig. 2: while the 2% and 5% data points are within error and not statistically different, the overlayed non-linear curve fit indicates that crosslinking does improve beyond 2% loading.  explain or remove the curve fit.

·         Fig. 2, caption:  describe the best-fit lines.  how were they calculated/drawn?  what are their formula?

·         Fig. 2: it appears the maximum "solubility" is approx 80%, potentially due to the lack of crosslinking with glycerol.  in later discussion, glycerol is attributed as 'free'.  this requires further clarification and discussion

·         Fig. 2: strange increment of right y-axis. add 2 ticks between labels

·         Table 2: change “hardly” to “minimally”

·         3.2: -OH stretching is due to hydroxyls in PVA, glycerol, and amylose. 

·         3.2: assign strong broad 1100-1000 peak ether C-O

·         3.2, Fig. 3: explain lack of c=o ester peak with increasing glutaraldehyde concentration?

·         3.2, Fig. 3: explain the peaks at ~1300 and ~800 that appear only in b and d? what are they? Why only in b and d?

·         3.3, line 231: these results suggest only the PVA is crosslinked with glutaraldehyde. Explain

·         Table 3: why does crystallinity degree continues to decrease with increased glutaraldehyde while absorption and solubility "level off" at ~2%?

·         Page 8, line 250: 5 wt% and less crosslinking would not significantly decrease hydroxyl count.  at most it would decrease to 95%

·         Page 8, line 252: more likely that increased cross-linking reduced the capacity of the polymer to swell (increase its volume, thus limiting capacity)

·         Fig. 5: the data is incomprehensible in fig. 5. data too close together in overlay. cannot distinguish data plots.  separate into different plots, use different data lines, or zoom in on regions of interest

·         Fig. 5: are the data significantly different? perform statistical analysis (ANOVA).  error bars appear to all overlap.

·         Page 9, line 272: however in previous fig 5, it is shown that the g5 sample absorbed the most water at approx. 40wt%. therefore the g5 at 90% RH would be quite soft.  this contradicts your conclusion here.  revise or explain

·         Table 4: overall, fairly very improvements in tensile strength with cross-linking.  Compare to commercial food packaging or agriculture films for context

·         Table 4: elongation at break for APF/G0.25 at 25%RH is anomalously low. Explain or repeat.

·         3.6: very subjective discussion.  compare the fracture analysis to literature, known fracture mechanics that have been extensively correlated with crosslinking and crystallinity

Author Response

Dear reviewer, thank you for your advice.  We have made changes according to your suggestions.

Reviewer 2 Report

The article is very well written and easily comprehensible. I do have the following comments/suggestions which I believe might help add more value to the manuscript:

1. Can the cross-linking mechanism be represented by a chemical equation? If so, please include that.

2. SEM micrographs of the control and the amylose/PLA/GLU samples showing the formation of pores might augment the conclusions on crosslinking. For instance, a higher degree of cross-linking would result in fewer pores with a smaller diameter and vice versa. It might be helpful to include the images of the pore network for each of the test samples to see how they compare and whether the observations are in agreement with the SD calculated in each case.

3. Do a thorough grammar check and ensure that the formatting is consistent.

Author Response

(The authors gave the same response as above.)

Reviewer 3 Report

This manuscript needs much improvement in many sections prior to the publications. There are lot of grammar mistake. Manuscript is not written well.

Specific comments.

In title, first character of each word should be capitalized.

References and citations are wrong as per the journal guidelines. Please check journal guideline or previous publish paper of polymers journal.

In abstract background sentence is missing! Please include it.

FTIR discussion need to rewrite and make characteristics bands in a tabular format.

Introduction need to be rewritten why crosslinking is imp for enhancing mechanical properties.

Why PVA have poor mechanical properties.

What the limitations of this study.

Table composition is incorrect. Comment about it. How amylose content 100wt%.

Crystallinity degree need to present in the table? Which one is 100 % than how authors calculate relative crystallinity please indicate clearly in the manuscript.

In affiliation section please mark ‘a’ or * as co first authors it should be marked after 1 not before author name.

Few references are suggested, cite it accordingly in the manuscript Carbohydrate Polymers 257 (2021) 117633; https://doi.org/10.1007/s10924-022-02454-w; https://doi.org/10.1007/s00289-009-0158-4; https://doi.org/10.1016/j.jiec.2018.07.029; Food packaging and Shelf Life, 33 (2022) 100904; progress in organic coatings 173 (2022) 107286. 

Please compare your study with other recent study preferable in a tabular format?

All the tabular data need statistical analysis. Mention also SD?

Where is Stress-strain curve? It is necessary to explain the mechanical properties.

Scale bar need to marked clearly in SEM images.

Insert the citations for the conclusion of the FTIR part and also in characteristics band.

In FTIR there is no indication of OH peak intensity was decreased after crosslinking. Authors need to highlight this part inside the figure.

Mention all the chemical and instrument details such model no, city, country etc.

Make space number and oC in line 148. Check these types of errors throughout the manuscript.

In this paper, crosslinking effect is more important so please calculate the crosslinking effect in quantitively manner. Please follow this paper progress in organic coatings 173 (2022) 107286. 

Lot of grammar mistake. Please correct it.

Conclusions section need to be rewritten preciously.

Insert quantitative data.

There are no citations in section 3.4.

Remove dot mark in wt.% for ex: in fig. 2 x-axis caption. Check other places of manuscript.

Insert citations to support your results of cross linking, section 3.1.

In all equation mention the % symbol in left side and ×100 in right side. For example: write equation as MC (%) = (w2-w1)/w1×100.

In all equations please provide citations.

Line 150, check 1% wt?

Line 167, no need another paragraph.

Authors stated this film can applicable in s agricultural mulch films therefore thermal stability need to be determined. Please provide TGA data.

Author Response

(The authors gave the same response as above.)

Round 2

Reviewer 3 Report

The authors improved the manuscript partially.

1. Authors mentioned few results are included in the supplementary file but I don't find any supplementary file.

2. Another problem; All data are included but not cited in the main manuscript. For example, Table S1 or S2 or S3 are not indicated in the main manuscript. Please revise carefully before publications your manuscript. How Table S2 will come two times. Please check your cover letter.

Table S2 Relative crystallinity degree of the blend films

Table S2. Comparison among APF Gx and commercial products

3. Crystallinity degree need to present in the table? Which one is 100 % than how authors calculate relative crystallinity. please indicate clearly in the manuscript.

Response: Thanks for your advice, WAXD diffraction patterns were fitted to measure crystallinity by peak segmentation. Similar crystallinity measurement methods were referred to the following reference and calculated by MDI jade 5.0. We have indicated the corresponding crystallinity in the revised manuscript and Table S2.

My question which one is 100%. You are calculating relative crystallinity %. One sample is 100%. In my opinion, PVA is 100%. other should calculate relatively to PVA.

4. References cited in the text are wrong. Polymers journal need [1]…..2…. For ex: line 81:

The mechanical properties of the starch-based 81 film were improved with glycerol as a plasticizer and polyvinyl alcohol (PVA) as a second component [22-24]. Not in superscript.

4. “Remove dot mark in wt.% for ex: in fig. 2 x-axis caption. Check other places of manuscript. Insert citations to support your results of cross linking, section 3.1.

Response: Thanks for your advice, the article has been revised.”

Dot mark still in the manuscript in many places including figures 2 and 5 and please check whole manuscript.

5. Few references are suggested, cite it accordingly in the manuscript Carbohydrate Polymers 257 (2021) 117633; https://doi.org/10.1007/s10924-022-02454-w; https://doi.org/10.1007/s00289-009-0158-4; https://doi.org/10.1016/j.jiec.2018.07.029; Food packaging and Shelf Life, 33 (2022) 100904; progress in organic coatings 173 (2022) 107286.

All references are not cited in the manuscript. Please check it again carefully.

 6.  Fog. 3 Y- axis transmittance unit is missing. it should be Transmittance (a.u.)

Author Response

Dear Editor and Reviewers,

Thank you very much for reviewing our manuscript. We are indeed greatly impressed by the conscientious manner and scientific advice of the reviewer. Now we have revised the manuscript according to the reviewers’ comments. All the changes made are highlighted in red color in the revised manuscript. A list of the detailed point-by-point responses to the reviewer’ comments are enclosed as follows.

  1. Authors mentioned few results are included in the supplementary file but I don't find any supplementary file.

Thank you for your suggestion. Our manuscript contains no supplementary information. The supplementary materials provided later, such as Figure S1/S2, are only for you to answer your valuable suggestions on our manuscript.

  1. Another problem; All data are included but not cited in the main manuscript. For example, Table S1 or S2 or S3 are not indicated in the main manuscript. Please revise carefully before publications your manuscript. How Table S2 will come two times. Please check your cover letter.

Table S2 Relative crystallinity degree of the blend films

Table S2. Comparison among APF Gx and commercial products

Sorry, we will carefully check the annotation of the picture.

  1. Crystallinity degree need to present in the table? Which one is 100 % than how authors calculate relative crystallinity. please indicate clearly in the manuscript.

Response: Thanks for your advice, WAXD diffraction patterns were fitted to measure crystallinity by peak segmentation. Similar crystallinity measurement methods were referred to the following reference and calculated by MDI jade 5.0. We have indicated the corresponding crystallinity in the revised manuscript and Table S2.

My question which one is 100%. You are calculating relative crystallinity %. One sample is 100%. In my opinion, PVA is 100%. other should calculate relatively to PVA.

Sorry, please forgive me for not understanding your meaning before. The crystallinity in this paper is calculated according to the ratio of the integral area of the sample to the calculated area of the fully crystallized PVA. The crystallinity of PVA used in this paper is not completely crystalline. Therefore, the crystallinity of PVA used in this paper cannot be considered as 100%.

  1. References cited in the text are wrong. Polymers journal need [1]…..2…. For ex: line 81:

The mechanical properties of the starch-based 81 film were improved with glycerol as a plasticizer and polyvinyl alcohol (PVA) as a second component [22-24]. Not in superscript.

Thank you for your suggestion. We have changed the format of references in the manuscript to a unified format.

  1. “Remove dot mark in wt.% for ex: in fig. 2 x-axis caption. Check other places of manuscript. Insert citations to support your results of cross linking, section 3.1.

Response: Thanks for your advice, the article has been revised.”

Dot mark still in the manuscript in many places including figures 2 and 5 and please check whole manuscript.

Thank you for your suggestion. We have re-examined the whole manuscript. In section 3.1, we have inserted the relevant citations.

  1. Few references are suggested, cite it accordingly in the manuscript Carbohydrate Polymers 257 (2021) 117633; https://doi.org/10.1007/s10924-022-02454-w; https://doi.org/10.1007/s00289-009-0158-4; https://doi.org/10.1016/j.jiec.2018.07.029; Food packaging and Shelf Life, 33 (2022) 100904; progress in organic coatings 173 (2022) 107286.

All references are not cited in the manuscript. Please check it again carefully.

Thank you for your suggestion. We have quoted all your recommendations as articles, but the progress in organic coatings 173 (2022) 107286 you recommended was not retrieved in the public database, so it was not quoted. I am very sorry.

  1. Fog. 3 Y- axis transmittance unit is missing. it should be Transmittance (a.u.)

Sorry, this error has been corrected.

Best regards,

Xinqing Wang  Zhenhua Huang

Prof. Fangping Chen and Prof. Changsheng Liu

Round 3

Reviewer 3 Report

The authors improved the manuscript.